# The Data Use Ontology to streamline responsible access to human biomedical datasets

## Graphical abstract

## Highlights

- Biomedical advances depend on the efficient and compliant re-use of sensitive human data

- The Data Use Ontology standardizes terms and definitions for consented data uses

- The Data Use Ontology facilitates discovery of, request for, and access to datasets

- Over 200,000 datasets worldwide have been annotated using the Data Use Ontology

## Authors

Jonathan Lawson, Moran N. Cabili, Giselle Kerry, ..., Jaime M. Guidry Auvil, Tommi H. Nyrönen, Mélanie Courtot

## Correspondence

mcourtot@gmail.com

## In brief

The GA4GH Data Use Ontology (DUO) provides unambiguous, machine-readable standard language for consent forms and the data sharing policies they represent. Lawson et al. describe the DUO standard and implementations throughout the data access workflow to expedite data access while maintaining or improving compliant processes.

Lawson et al., 2021, Cell Genomics *1*, 100028
November 10, 2021 © 2021 The Author(s).

# Cell Genomics

CellPress

## Technology

# The Data Use Ontology to streamline responsible access to human biomedical datasets

Jonathan Lawson,[1,37] Moran N. Cabili,[1,37] Giselle Kerry,[2] Tiffany Boughtwood,[3] Adrian Thorogood,[4,5] Pinar Alper,[5] Sarion R. Bowers,[6] Rebecca R. Boyles,[7] Anthony J. Brookes,[8] Matthew Brush,[9] Tony Burdett,[2] Hayley Clissold,[6] Stacey Donnelly,[1] Stephanie O.M. Dyke,[10] Mallory A. Freeberg,[2] Melissa A. Haendel,[9] Chihiro Hata,[11] Petr Holub,[12] Francis Jeanson,[13] Aina Jene,[14] Minae Kawashima,[15] Shuichi Kawashima,[16] Melissa Konopko,[17] Irene Kyomugisha,[18] Haoyuan Li,[19] Mikael Linden,[20] Laura Lyman Rodriguez,[21] Mizuki Morita,[22] Nicola Mulder,[23] Jean Muller,[24,25]

*(Author list continued on next page)*

[1]Broad Institute of Harvard and the Massachusetts Institute of Technology, Cambridge, MA, USA
[2]European Molecular Biology Laboratory—European Bioinformatics Institute (EMBL-EBI), Hinxton, UK
[3]Australian Genomics, Murdoch Children's Research Institute, Parkville, VIC, Australia
[4]Centre of Genomics and Policy, Department of Human Genetics, McGill University, Montreal, QC, Canada
[5]ELIXIR-Luxembourg, Luxembourg Centre for Systems Biomedicine, University of Luxembourg, Belvaux, Luxembourg
[6]Wellcome Sanger Institute, Wellcome Genome Campus, Hinxton, UK
[7]RTI International, Research Triangle Park, NC, USA
[8]University of Leicester, Leicester, UK
[9]University of Colorado Anschutz Medical Campus, Aurora, CO, USA
[10]McGill Centre for Integrative Neuroscience, Montreal Neurological Institute, Department of Neurology & Neurosurgery, Faculty of Medicine, McGill University, Montreal, QC, Canada
[11]Bioinformation and DDBJ Center, National Institute of Genetics, Mishima, Japan
[12]BBMRI-ERIC, AT and Masaryk University, Brno, Czech Republic
[13]University Health Network, Toronto, ON, Canada
[14]Centre de Regulació Genòmica (CRG), Barcelona, Spain
[15]National Bioscience Database Center, Japan Science and Technology Agency, Tokyo, Japan
[16]Database Center for Life Science, Joint Support-Center for Data Science Research, Research Organization of Information and Systems, Kashiwa, Japan

*(Affiliations continued on next page)*

## SUMMARY

Human biomedical datasets that are critical for research and clinical studies to benefit human health also often contain sensitive or potentially identifying information of individual participants. Thus, care must be taken when they are processed and made available to comply with ethical and regulatory frameworks and informed consent data conditions. To enable and streamline data access for these biomedical datasets, the Global Alliance for Genomics and Health (GA4GH) Data Use and Researcher Identities (DURI) work stream developed and approved the Data Use Ontology (DUO) standard. DUO is a hierarchical vocabulary of human and machine-readable data use terms that consistently and unambiguously represents a dataset's allowable data uses. DUO has been implemented by major international stakeholders such as the Broad and Sanger Institutes and is currently used in annotation of over 200,000 datasets worldwide. Using DUO in data management and access facilitates researchers' discovery and access of relevant datasets. DUO annotations increase the FAIRness of datasets and support data linkages using common data use profiles when integrating the data for secondary analyses. DUO is implemented in the Web Ontology Language (OWL) and, to increase community awareness and engagement, hosted in an open, centralized GitHub repository. DUO, together with the GA4GH Passport standard, offers a new, efficient, and streamlined data authorization and access framework that has enabled increased sharing of biomedical datasets worldwide.

## INTRODUCTION

To address global scientific challenges in health, human biomedical data must be shared and integrated worldwide.[1] To promote discovery and improve healthcare, researchers and clinicians need to be able to find, access, harmonize, and re-use data from diverse data sources. Data access for research is often facilitated by data repositories, and in a growing number of federated data environments[2] that aggregate datasets within or among themselves and make the results available to the

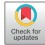

**CellPress**

**Cell Genomics**
Technology

Satoshi Nagaie,[26] Jamal Nasir,[27] Soichi Ogishima,[26] Vivian Ota Wang,[28] Laura D. Paglione,[29] Ravi N. Pandya,[30] Helen Parkinson,[2] Anthony A. Philippakis,[1] Fabian Prasser,[31] Jordi Rambla,[14] Kathy Reinold,[1] Gregory A. Rushton,[1] Andrea Saltzman,[1] Gary Saunders,[17] Heidi J. Sofia,[32] John D. Spalding,[2] Morris A. Swertz,[33] Ilia Tulchinsky,[34] Esther J. van Enckevort,[33] Susheel Varma,[35] Craig Voisin,[34] Natsuko Yamamoto,[36] Chisato Yamasaki,[36] Lyndon Zass,[23] Jaime M. Guidry Auvil,[28] Tommi H. Nyrönen,[20] and Mélanie Courtot[2,38,*]

[17]ELIXIR Hub, Wellcome Genome Campus, Hinxton, UK
[18]Division of Human Genetics, Faculty of Health Sciences, University of Cape Town, Cape Town, South Africa
[19]Canada's Michael Smith Genome Sciences Centre, Vancouver, BC, Canada
[20]ELIXIR-Finland, CSC - IT Center for Science Ltd, Espoo, Finland
[21]Patient-Centered Outcomes Research Institute, Washington, DC, USA
[22]Okayama University, Okayama, Japan
[23]Computational Biology Division, IDM, Faculty of Health Sciences, University of Cape Town, Cape Town, South Africa
[24]Laboratoire de Génétique Médicale, Institut de Génétique Médicale d'Alsace, INSERM U1112, Université; de Strasbourg, Strasbourg, France
[25]Laboratoire de Diagnostic Génétique, Institut de Génétique Médicale d'Alsace, Hôpitaux Universitaires de Strasbourg, Strasbourg, France
[26]Tohoku Medical Megabank Organization (ToMMo), Tohoku University, Sendai, Japan
[27]Department of Life Sciences, University of Northampton, Northampton, UK
[28]Office of Data Sharing, National Cancer Institute, NIH, Rockville, MD, USA
[29]Spherical Cow Group, Rego Park, NY 11374, USA
[30]Microsoft Research, Redmond, WA 98052, USA
[31]Berlin Institute of Health at Charité—Universitätsmedizin Berlin, Berlin, Germany
[32]National Human Genome Research Institute, NIH, Bethesda, MD, USA
[33]Genomics Coordination Center, Department of Genetics, University of Groningen, University Medical Center Groningen, Groningen, the Netherlands
[34]Google Cloud, Kitchener, ON N2H 5G5, Canada
[35]Health Data Research UK, Gibbs Building, 215 Euston Road, London NW1 2BE, UK
[36]Osaka University, Osaka, Japan
[37]These authors contributed equally
[38]Lead contact
*Correspondence: mcourtot@gmail.com

research community. Challenges arise in the aggregation of datasets with varying ethical or regulatory conditions on data reuse. Different conditions may stem from different applicable data protection laws (e.g., limits on allowable purposes of processing, transfers to third countries), informed consents (e.g., specific vs. broad), policies (e.g., IRB data release authorizations), or data sharing agreements (e.g., within consortia).[3] Due to this heterogeneity of re-use conditions, it can be difficult for researchers to search and find appropriate datasets, methods of requesting and accessing those datasets vary, and there is no shared understanding of the allowable uses and/or downstream analyses of the data once access is approved.

Current processes to access sensitive human biomedical data can be cumbersome, time and cost intensive, and variable between repositories. In typical workflows, Data Access Committees (DACs) manually review data use terms; this process can be delayed by the need to interpret data use terms often described in inconsistent and ambiguous language. There can also be inconsistency in access determinations across DACs, particularly for broadly defined data use terms, such as "permitted use for a disease and related conditions." Similarly, language in a consent form prohibiting "commercial use" has been interpreted differently by DACs, ranging from not allowing commercial organizations access to the data to not allowing the data to be used for commercial purposes—independently of the organization type. Finally, these interpretations can shift over time, increasing the risk that data are used in a way that does not reflect what the research participant originally agreed to and leading to inconsistent data sharing practices.

To address the needs for consistent terminology and reliable interpretations of allowable data uses, the GA4GH Data Use and Researcher Identities (DURI) work stream[4] developed a data authorization and access framework to streamline the process for granting researchers access to biomedical datasets based on their credentials and research purposes. A main component of this framework is the Data Use Ontology (DUO), a standard, machine-readable vocabulary of data use terms that enables direct matching between data use conditions and intended research use. DUO is complemented by the GA4GH Passport standard (see Voisin et al. in this issue),[5] which provides a machine-readable representation of a researcher's data access permissions. Together, the GA4GH DUO and Passport standards enable automating access by researchers to multiple datasets based on their authentication and authorization levels and has been deployed by various organizational members of the GA4GH DURI work stream. DUO is now the accepted GA4GH standard for data use terms, based on use cases from several GA4GH Driver Projects.[6] Australian Genomics, EGA, GEnome Medical alliance (GEM) Japan, Human Heredity & Health in Africa (H3Africa), U.S. National Heart, Lung, and Blood Institute, BBMRI-ERIC, and U.S. National Cancer Institute have all contributed to the establishment and review of DUO terms, which are aligned with data use terms or phrasing of their respective consent forms. Over 200,000 datasets worldwide have been annotated with

**Table 1. Count of datasets annotated with DUO by data custodian as of February 2021**

| Data custodian | Datasets annotated with DUO |
|---|---|
| Broad Institute | 225 |
| Sanger Institute | 700 |
| EGA | 1,021 |
| HDR UK | 568 |
| BBMRI-ERIC | In progress. Manual for data managers with guidance for DUO annotations released: https://doi.org/10.5281/zenodo.4427731 |
| AMED Biobank Network (GEM Japan) | 203,900 |
| Australian Genomics | 14 |
| H3Africa | 16 |

A census of datasets annotated with DUO in February 2021 highlights widespread adoption of the standard. Early implementers such as EGA are now requiring DUO annotation upon dataset submission. New partners such as BBMRI-ERIC are only starting the annotation process. AMED Biobank has made a very large number of DUO annotations, as they consider each sample to be its own dataset. An example implementation in the EGA is described in supplemental information.

machine-readable DUO terms (Table 1). DUO has been successfully leveraged by software such as the Broad Data Use Oversight System (DUOS) to enable automated matching between access requests and DUO annotation on datasets (see Cabili et al. in this issue).[7]

In this study, we report on the DUO standard, describe the curated structured vocabulary and hierarchies, and review use cases and considerations in implementing DUO for the management and access of biomedical datasets. DUO has been successfully used to annotate genomics datasets worldwide, and its usage is being expanded to direct mapping into consent forms and automated matching of requests to permissions by DACs. Future uses of DUO include annotation to different data types such as samples and integration within GA4GH Passport visas.

## DESIGN

DUO is a structured vocabulary of standard human- and machine-readable data use terms. DUO's original list of terms was informed by review of common terminologies used by major international controlled-access genomic repositories (e.g., U.S. National Institutes of Health database for Genotypes and Phenotypes, NIH dbGaP,[8] and European Genome-Phenome Archive, EGA[9]), as well as policy tools developed by the GA4GH Regulatory and Ethics Work Stream (REWS).[3,10] Contributors from those efforts joined to form the Data Use group, which met regularly both through videoconferences and face-to-face meetings. External efforts such as the Informed Consent Ontology (ICO)[11] were additionally reviewed for interoperability and synergistic evolution; DUO has been directly imported in ICO to describe data use conditions instead of duplicating its content. The DUO terms are intended to be a

simple set of data use terms most often used or referenced in consent forms that include provisions for data sharing. DUO does not aim to represent all possible data use terms, consent phrases, or complex logical permutations of permissions, limitations, or requirements. Structurally, DUO contains 25 terms representing two types of data use terms, permissions and modifiers (Table S1):

- Permission terms include "general research use," "health or medical or biomedical use," "disease specific research," and "population origins or ancestry research only" and are expressly permitted uses or focused areas of research.
- Modifier terms add requirements, limitations, or prohibitions within the permitted boundary (Figure 1).

DUO is use-case driven, and requests for new data use terms in DUO must be supported by specific use cases that promote and facilitate data sharing. Each DUO term was developed based on contributions and reviews from community experts and implementers. Contributions to DUO are public and created by raising GitHub issues;[12] anyone may submit a request to add a new term or comment on an existing request. Requests are discussed by the DUO work stream leads and driver project implementers on the tracker, on the DUO mailing list, and during periodic teleconferences. Once approved, changes are open to the public for further discussion over a comment period of 2 weeks, as per the DUO governance policy.[13]

DUO is implemented in the Web Ontology Language (OWL),[14] a World Wide Web Consortium standard. Development of DUO follows Open Biomedical Ontologies (OBO) development principles,[15] ensuring interoperability with other ontological resources, such as those describing disease entities.[16] As per OBO guidelines, DUO is built under the Basic Formal Ontology (BFO)[17] upper-level ontology. The DUO root terms "data use permission" and "data use modifier" are subclasses of "data item" (IAO:0000027), itself a type of "information artifact entity" (IAO:0000030) and "generically dependent continuant" (BFO:0000031). While BFO provides the framework for the DUO hierarchy, it proved confusing to use for most users. We consequently worked with the developers of the EMBL-EBI Ontology Lookup Service (OLS)[18] to design and implement a system allowing selection of suitable entry levels in the DUO hierarchy. The "preferred root" toggle shown in Figure 2 allows most users to browse only classes of interest, while expert ontologists can instead select the complete view. DUO terms are stable, with each DUO term having its unique Uniform Resource Identifier, which can be browsed using the OLS. Most importantly, the meaning associated with a specific DUO ID is permanent; this guarantees consistency through time of the data use terms. Different versions of DUO are available through the GitHub repository,[19] including an editors' version that captures ongoing development and stable, released versions. Released versions of DUO are associated with permanent URLs (PURLs) for sustainability:[20] the most recent release is always available from http://purl.obolibrary.org/obo/duo.owl, while previous versions can be accessed through their date-based PURL, providing choice for users who prefer to use a specific historical view of the ontology[21,22] for stability while transitioning to the latest version.

Figure 1. Data Use Ontology permissions and modifiers

DUO is a hierarchical vocabulary of data use terms most often used to denote secondary usage conditions for controlled access datasets. DUO does not aim to represent all possible data use terms, consent phrases, or complex logical permutations of permissions, limitations, or requirements. As of June 2021, DUO contains 25 terms representing two types of data use terms, permissions and modifiers. Permissions such as General Research Use (GRU), Health or Medical or Biomedical use (HMB), Disease Specific research (DS), and Population Origins and Ancestry research (POA) standardize allowed usage of the datasets. Modifiers are used to further qualify main categories of controlled access.

Terms are positioned in the DUO hierarchy, such that subclasses are more specific sets of instances than their parents. This allows for inference of new knowledge through description logic underpinning OWL reasoners.[23] For example, when searching for datasets for a "disease-specific" research use (Figure 2), a researcher would see query results of datasets matching this use term and its parents, "health and biomedical research" (direct superclass) and "general research" (indirect superclass). The initial structure of the repository was generated using the ontology development kit,[24] which provides a way of creating an ontology project ready for pushing to GitHub. Development of the ontology follows a modular approach for greater flexibility both by developers of DUO and its users. For example, the DUO Japanese translation is stored as a separate file from the main ontology. This file is merged in at release time via an automated script, allowing different files and features to remain independent until they are ready to be published and/or to be excluded at release time on demand—for example, for users who do not require translations from English. The same script also executes SPARQL[25] queries to render CSV versions, again for easy human browsing in the GitHub repository. Finally, the script merges relevant subsets of external ontologies imported through the MIREOT method[26] to promote ontology re-use and consistent identification of ontology terms across resources.

To increase community awareness and engagement, DUO is hosted under an open, centralized GitHub repository. This enables tagging of versions and continuous integration tests to be run at each iteration via the Travis CI software. After each modification of the source file, the ELK reasoner[27] is run to ensure ongoing consistency of the ontology.

## RESULTS

To ensure trustworthiness and sustainability of its technical standards, the GA4GH applies an open and consistent development and product approval process.[1] In 2019, DUO was unanimously approved as a GA4GH standard by the GA4GH Steering Committee, joining other products in the GA4GH Genomic Toolkit suite.[1] Figure 3 displays the current implementers of DUO.

DUO has been incorporated in several central aspects of the data access request process (Box 1). First, DUO terms are applied as dataset metadata to be stored alongside the data they describe in a repository, making it easier for data custodians to manage their datasets compliantly and facilitate researchers' querying of the datasets by their data use terms. Repositories can add DUO annotations to their dataset files, either retrospectively through curation of existing data or interactively at submission time. Users can search for datasets according to data use terms to determine what datasets are available for their purposes before requesting data access. This improved accessibility and interoperability of datasets increases their FAIRness:[28] 2.6% of data requesters who applied for access to Sanger's Cancer Genome Project (CGP) datasets between April and October 2020 had used the EGA DUO search tool to find re-usable datasets compatible with their research purposes.

In a second use case, DUO terms have been leveraged by DACs to facilitate and, for the first time, automate parts of the

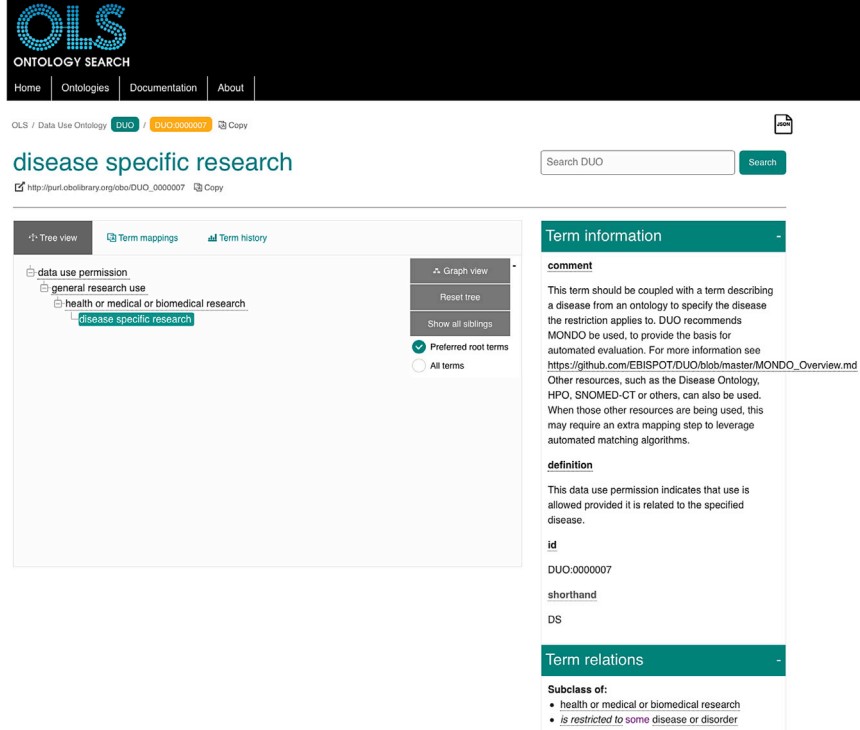

**Figure 2. Browsing the Data Use Ontology**
The DUO OWL file has been loaded in human-friendly browsers such as the Ontology Lookup Service (OLS). This enables interactive navigation through the hierarchy and display of additional properties such as definition, comment, or relations to other terms. For example, the "disease specific research" DUO term, http://purl. obolibrary.org/obo/DUO_0000007, clarifies that it should be used in conjunction with a term from a disease ontology. The "Preferred root terms" button (middle, active green checkbox) guides display of the top classes to be displayed to the user instead of presenting the complex upper-level BFO hierarchy (accessible by selecting "All terms")

data access request process. The use of DUO in electronic data access systems enables automated matching by software algorithm, leveraging the DUO hierarchy and logical structure. An implementation in automating data access requests has been piloted for NIH and the Broad Institute through DUOS[7] and is now being extended to other databases. The DUOS software platform performs automated DUO-based data use oversight and provides interfaces to simplify the work of DACs. An empirical evaluation of the results demonstrates that the DUO is broadly useful, matching ~96% of consent terms in examined datasets, and that using DUOS to automate the process streamlines the review process while maintaining efficacy and consistency.

As a third use case, DUO terms are incorporated into the data sharing language in consent forms written during the study inception.[30,31] Incorporating DUO terms at this early stage is important to enable more effective and consistent data use management. This addresses current challenges in the common use of informed consent language that does not fully capture the scope and issues related to data sharing and secondary research purposes, resulting in uncertainty for participants regarding research expectations as well as for data providers and data stewards or DACs in assessing how datasets can be distributed. The consent clauses in the Machine-Readable Consent Guidance are accompanied by explanations and guidance for consistency, and to ensure prospective capture as machine-readable data use terms. This is currently undergoing evaluation and validation by IRBs, and we anticipate this becoming a recommendation that could be more broadly followed.

## DISCUSSION

Since its approval as a GA4GH standard,[1] DUO has been widely implemented across diverse biomedical projects worldwide. Beyond requests for and comments on new data use terms, DUO standard implementers have contributed by proposing translations in other languages, such as Japanese, or in "plain language," which has been shown to increase understanding and participation of research participants.[32] To this end, DUO was successfully extended for consent use as the Machine-Readable Consent Guidance described earlier, which was approved as a GA4GH standard in July 2020[33] and is being actively reviewed and implemented by IRBs and research studies. In addition, community members enthused by the success and simplicity of DUO aim to further extend its application beyond genomic datasets to resources such as biological specimens, imaging data, and public health data. The Finnish Institute for Health and Welfare biobank[34] has already implemented DUO in requiring sample depositors to describe sample/data use terms when depositing in their repositories. Indeed, nothing precludes developing applications or extensions of DUO for other scientific resources. Successful external extensions of the standard can be fed back to GA4GH, allowing for continual improvement in utility and function for the community.

DUO terms can also be used in healthcare settings and alongside complementary standards. Health Level Seven International (HL7)'s Fast Healthcare Interoperability Resources (FHIR)[35] Consent resource,[36] as well as other tools or standards, such as the Automatable Discovery and Access Matrix (ADA-M) or OASIS's LegalRuleML,[37] use logic for expressing more complex data use rules. The HL7 standard permits an implementer to adopt a default rule for a given use term (e.g., everything permitted by default, everything restricted by default) and then specify exceptions. LegalRuleML and ADA-M explicitly define if a rule for coded data use is a permission, prohibition, or condition. This approach requires users to "translate" their intuitive thinking into machine-based logic and can lead to complexity, confusion, and a greater risk of error.

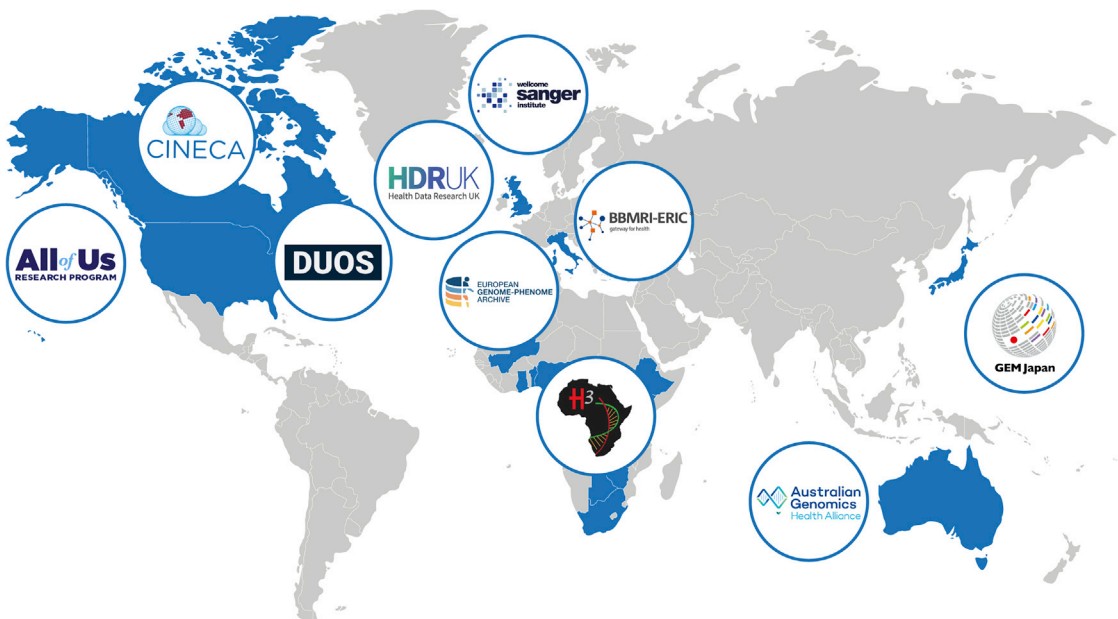

**Figure 3. Current implementations of the Data Use Ontology**
DUO has been implemented to annotate genomics datasets worldwide. As of November 2021, implementers include repositories, databases, and projects in North America, Europe, Africa, Europe, Asia, and Australia.

### Limitations of the study

The GA4GH DUO standard represents the data use terms commonly used by data management professionals for sharing of biomedical datasets, while minimizing the complexity of logical permutations of data use terms, essential to global interoperability and data sharing.[38] For example, DUO adopts the term "not-for-profit use only" rather than decomposing "profit" and whether it is "allowed," "forbidden," or "restricted" in specific instances, thus not requiring users to mix and match terms with potentially opposing meanings; DUO is not built to capture the entire spectrum of possible data use combinations, as pursuing a vocabulary to describe all possible combinations of data use would likely lead to an infinitely complexifying model given the constant increase of possible terms and combination permutations. This intentional limitation of the DUO terminology space has been encouraged by researchers, in line with the DURI leadership's vision for DUO as a concise standard to facilitate compatibility of terms.

Arguments to the contrary espouse DUO and the aspiration for a limited vocabulary as counter to the needs of specific participant communities. A red herring example often used to justify this contrary position is that rare disease research participants often believe that DUO's limited scope would not be able to represent the unique, specific diseases they have, such as ataxia-telangiectasia or Diamond-Blackfan anemia. Yet this reflects an inversion of understanding, as permitting unique, edge-case-like types of research would be permissible via many of the existing DUO terms, particularly those such as General Research Use and Health/Medical/Biomedical Use. Annotating those datasets with more general DUO terms also increases the probability of researchers reaching those disease-specific findings, possibly impacting scientific discoveries to prevent and treat such diseases. Ultimately, after engaging with the DUO team, representatives of the RARE-X rare disease community became strong proponents of DUO and advocate for its use among other rare disease participant groups. To help clarify this to future adopters of DUO, the DURI work stream is actively developing DUO implementation guidance and is also evaluating whether it would be feasible to provide a DUO-based software service to aid groups in choosing DUO terms that fit their needs.

Currently, the implementation and use of DUO may be limited by the need to retrospectively translate consent form language into DUO terms. This limits the number of dataset annotations possible and potentially generates variability in the mapping of legacy consent form conditions to DUO terms. To prospectively mitigate this issue, we have finalized the Machine-Readable Consent Guidance[29] to propose a consent form already mapped into DUO terms. DUO also supports DACs and data custodians with workshops and trainings on how to translate consent forms to DUO terms.

### Conclusion

DUO has been adopted worldwide for use in annotation of over 200,000 datasets to describe data use conditions for human biomedical data (Table 1). The GA4GH DUO and Passport standards, part of a joint strategy to streamline access to data, have not yet been connected to enable a singular process. As a next step, the DURI working group of GA4GH is planning to integrate DUO terms into Passport visas, combined with advocating for policy shift in approving access to groups of datasets by data use profile rather than individualized datasets. This will allow

**Box 1. DUO at each step of the data access process**

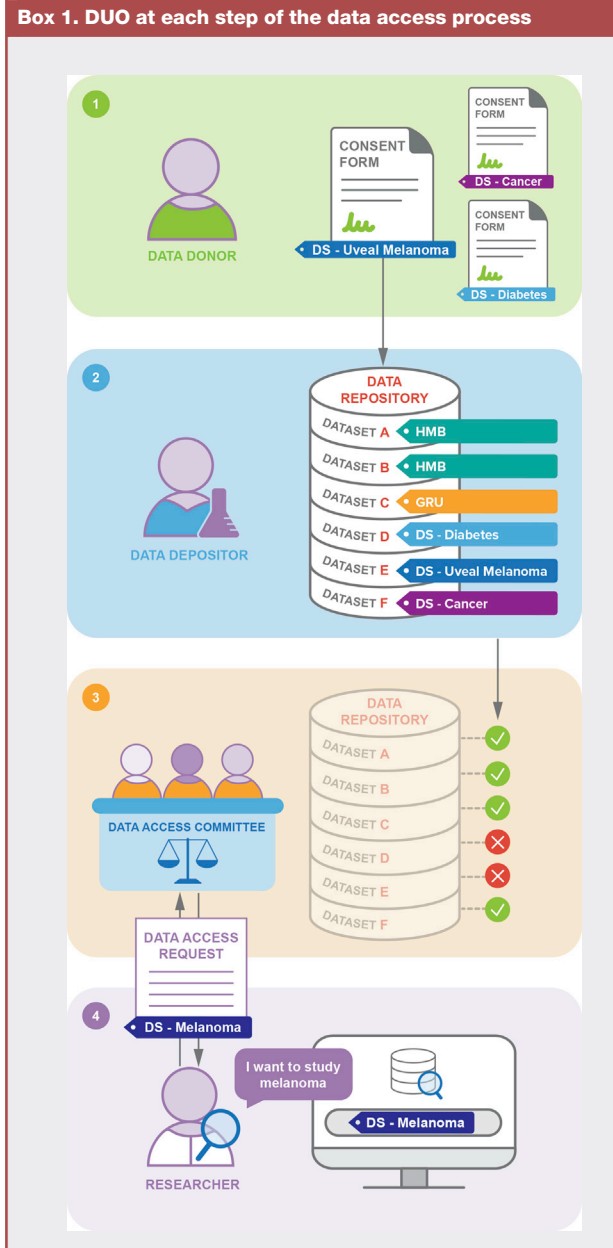

### STEP 1: CONSENT FORM ANNOTATION

Data donors—participants in trials and studies—agree to data use purposes described in consent forms. Consent forms are written by research teams in compliance with national, local, or institutional regulations and/or policies. To maintain stewardship and accessibility, these forms should adopt clear-language data use terms, and templates should be made publicly accessible. DUO standard data use terms can be embedded directly in the consent forms' clauses, following the GA4GH Machine-Readable Consent Guidance.[29] Organizations may add additional usage parame-

ters beyond DUO, for example, to protect intellectual property.

### STEP 2: DATASET ANNOTATION

Datasets hosted in controlled-access repositories are annotated with DUO terms denoting the data use terms that must be adhered to for approval for secondary data usage. The DUO terms can be added retrospectively by repository custodians for legacy datasets and/or prospectively by data depositors upon data submission.

### STEP 3: DATASET DISCOVERY

A researcher can use DUO terms to search for datasets with relevant use conditions in a data repository. For example, they can search for all datasets consented for melanoma research. This returns only the list of datasets that would be permitted for use given this specific condition. Alternatively, the researcher can query a specific dataset for their use case, without needing to contact the DAC or other help resources. This process allows the researcher to streamline the process of identifying suitable datasets and avoid unnecessary data access request submissions.

### STEP 4: DATA ACCESS REQUEST

A researcher requests access to relevant datasets and describes the research purpose using DUO terms. This enables efficient triaging by the DAC, either manually or using an automated matching algorithm.[7] The DAC reviews the access request to determine if the proposed research is consistent with the data use terms and if so, grants the researcher access to the datasets. The use of DUO terms facilitates a streamlined and standardized review by DACs.

authenticated researchers to automatically access new and existing datasets matching their DAC-approved data use profile after sign-in. Further streamlining the access process will minimize the need for multiple consecutive requests as new data are released either for a specific project or in a new repository. Such an approach also sets a precedent for establishing trust between DACs and enhanced alignment in the approval process: we envision users' data use profiles could be shared across DACs. As biomedical datasets are produced in greater numbers, across diverse settings, reliance on DUO-based mechanisms is critical to streamline data access to enable scientific collaborations.

### STAR★METHODS

Detailed methods are provided in the online version of this paper and include the following:

- KEY RESOURCES TABLE
- RESOURCE AVAILABILITY

## SUPPLEMENTAL INFORMATION

## ACKNOWLEDGMENTS

The authors would like to acknowledge the GA4GH Data Use and Researcher Identities (DURI) work stream. We are also grateful for contributions from the developers of the EMBL-EBI Ontology Lookup Service (Simon Jupp, Nico Matentzoglu, and Henriette Harmse) who implemented the "preferred root" visualization to support DUO users. Thanks to Stephanie Li for her ongoing help with graphics and design for this manuscript and other DUO materials. The DUO logo was designed by Spencer Philips at EMBL-EBI. We are grateful to Angela Page, Michael Baudis, and Peter Goodhand (GA4GH), Ewan Birney (EMBL, GA4GH), Bartha Knoppers and Anne-Marie Tassé (McGill University), Ayad Aliomer (Optum), Michele Mattioni (Seven Bridges), Neil Otte (University of Buffalo), and Cooper Stansbury (University of Michigan) for their support. M.A.S. and E.J.v.E. acknowledge contributions from Jelmer Veen, Marije van der Geest, and Aneas Hodselmans for BBMRI-NL Directory work.

G.K., M.A.F., H.P., J.D.S., and M.C. were funded by EMBL-EBI Core Funds and Wellcome Trust GA4GH award number 201535/Z/16/Z. T. Burdett was funded by EMBL-EBI Core Funds. T. Boughtwood was funded by NHMRC GNT111353, GNT200001, and the Australian MRFF. P.A. was funded by ELIXIR Luxembourg. S.D. and K.R. were funded by the Broad Institute. M.A.H. and M.B. received funding from NIH #5R24OD011883. M.L. and M.C. were funded by the CINECA project (H2020 No 825775). N.M. and L.Z. were funded by H3ABioNet, NIH grant number U24HG006941. S.O. and C.Y. received funding from the Japan Agency for Medical Research and Development (AMED) under grant numbers JP19kk020501 and JP18kk0205012. A.A.P. was funded by NHGRI AnVIL, award number U24HG010262. F.P. was supported, in part, by the European Union's Horizon 2020 research and innovation program under the EJP RD COFUND-EJP #825575. M.A.S. and E.J.v.E. were funded by FAIR genomes (ZonMW #846003201) and EOSC-Life (H2020 #824087). S.V. was funded by the Industry Strategy Challenge Fund by the UK Government. G.S. was funded by ELIXIR, the research infrastructure for life science data.

## AUTHOR CONTRIBUTIONS

All authors contributed to investigation and writing—review and editing. J.L., M.N.C., M.C., J.D.S., and A.A.P. contributed to conceptualization. J.L., G.A.R., A.A.P., S.D., A.S., L.L.R., T.H.N., M.C., M.N.C., A.J.B., F.J., S.O.M.D., S.R.B., H.C., G.K., and T. Boughtwood contributed to validation. G.K., T. Boughtwood, H.C., S.R.B., N.M., S.O., M. Kawashima, M.M., N.Y., S.N., C.H., S.K., A.J., J.R., J.D.S., M.A.S., E.J.v.E., S.V., C.Y., and L.Z. contributed to data curation. M.C., J.L., A.T., T. Boughtwood, G.K., P.A., S.R.B., H.C., P.H., H.P., F.P., J.R., A.J., G.S., J.D.S., M.A.S., S.V., C.V., S.O., M. Kawashima, N.M., and C.Y. contributed to writing—original draft.

## DECLARATION OF INTERESTS

M.N.C. is an employee of Foundation Medicine and equity holder of Roche. A.A.P. is a venture partner at GV and an employee of alphabet corporation. He has received funding from MSFT, Verily, IBM, Intel, Bayer, and Novartis. The views expressed by L.L.R. are the author's own and do not necessarily represent those of her organization.

## INCLUSION AND DIVERSITY

One or more of the authors of this paper self-identifies as an underrepresented ethnic minority in science. One or more of the authors of this paper self-identifies as a member of the LGBTQ+ community.

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

**Cell Genomics**
**Technology**

## STAR★METHODS

### KEY RESOURCES TABLE

| REAGENT or RESOURCE | SOURCE | IDENTIFIER |
|---|---|---|
| Software and algorithms | | |
| ELK reasoner | Kazakov et al., 2014[27] | https://www.korrekt.org/page/The_Incredible_ELK |
| Ontology Lookup Service | Jupp et al., 2015[18] | https://www.ebi.ac.uk/ols/index |
| Ontology Development Kit | https://douroucouli.wordpress.com/2018/08/06/new-version-of-ontology-development-kit-now-with-docker-support/ | https://doi.org/10.5281/zenodo.4662066 |
| DUO GitHub repository | This manuscript | http://purl.obolibrary.org/obo/duo |
| Released DUO file | This manuscript | http://purl.obolibrary.org/obo/duo.owl |

### RESOURCE AVAILABILITY

#### Lead contact
Further information and requests for resources should be directed to and will be fulfilled by the lead contact, Mélanie Courtot (mcourtot@gmail.com).

#### Materials availability
This study did not generate new unique reagents.

#### Data and code availability
The Data Use Ontology source files, scripts and documentation are licensed under CC-BY 4.0 and available from the GitHub repository http://purl.obolibrary.org/obo/duo. This manuscript describes the 2021-02-23 release of DUO, permanently publicly available at http://purl.obolibrary.org/obo/duo/releases/2021-02-23/duo.owl

### METHOD DETAILS

The GA4GH community has been previously involved in the development of two main controlled vocabularies/"information models" that systematically capture data use restrictions on human genomics and health datasets: (1) Consent Codes[10] and (2) ADA-M.[3] Further details on the process by which these vocabularies were created is described elsewhere.[3,10] Preceding these efforts, guidance from the NIH's database of Genotype and Phenotype (dbGaP)[8] led to the organic creation of a data use restriction vocabulary by requesting data depositors to represent the conditions for secondary use of the deposited datasets using the dbGaP vocabulary. This dbGaP vocabulary included a set of a handful of nucleating terms that are often used (such as: "General Research Use" (GRU), "Health/Medical/Biomedical research only" (HMB)) and also allowed depositors to add new terms to the vocabulary if a suitable term didn't previously exist.

The goal in creating DUO was to create a human and Machine-Readable representation of these 3 vocabularies and to code and maintain it in a form of a versioned ontology that will allow automated computation of software systems (e.g., as needed by a search function) on the ontology terms. An ontology encodes the hierarchy between terms which is critical for machine based automated computation. Before attempting to create DUO we defined 5 main goals:

1. Generate an ontology that is easy to use for the end user and unambiguous.
2. Generate a lean ontology based on real life use cases; and evolve gradually.
3. Ontology categories could be used to represent Data Use Conditions and Research Purposes. Thus, definitions should be generalized accordingly.
4. Include categories to support piloting ADA-M and Consent Codes as a human interface to define data use restrictions and research purposes.
5. Ideally, support a matching algorithm that uses boolean logic.

To create DUO we conducted the following steps:

1. Consent code and ADA-M integration proposal: In early 2018, we reviewed the Consent Codes,[10] the NIH dbGaP depositor guide and the ADA-M information model[3] and created a proposal of a set of data use restriction terms and their hierarchy as the basis for the DUO ontology.

2. DUO refinement: In the GA4GH 2018 spring in person meeting in Toronto we initiated a review process in which key potential DUO users trimmed down the set of terms to be included in the initial version of DUO and confirmed their hierarchy to ensure that common software-based use cases can be coded using DUO. These users included representatives from GA4GH driver projects (e.g, The All of Us research program, Australian Genomics, ANVIL), and representatives of data repositories that were seeking a Machine-Readable data use ontology (e.g, dbGaP, EGA, Sanger, The Broad Institute). These processes continued during the GA4GH bi-weekly DURI team video-conference meetings, where the team systematically discussed and approved terms, their definition and hierarchy in the ontology. Whenever a controversy arose the team relied on the guiding principles of creating (a) a lean ontology that (b) supports a real-life use case. In the absence of an immediate real life use case our team refrained from adding terms in favor of creating a lean ontology to begin with.

3. Ontology representation of DUO: Once a stable first version of DUO was agreed on, the ontology was implemented in the Web Ontology Language (OWL),[14] a World Wide Web Consortium standard. Development of DUO follows Open Biomedical Ontologies (OBO) development principles,[15] ensuring interoperability with other ontological resources, such as those describing disease entities.[16] As per OBO guidelines, DUO is built under the Basic Formal Ontology (BFO)[17] upper-level ontology. The DUO root terms "data use permission" and "data use modifier" are subclasses of "data item" (IAO:0000027), itself a type of "information artifact entity" (IAO:0000030) and "generically dependent continuant" (BFO:0000031). DUO terms are stable, with each DUO term having its unique Uniform Resource Identifier, which can be browsed using the OLS. Most importantly, the meaning associated with a specific DUO ID is permanent; this guarantees consistency through time of the data use terms. Different versions of DUO are available through the GitHub repository,[19] including an editors' version which captures ongoing development, and stable, released versions. Released versions of DUO are associated with permanent URLs (PURLs) for sustainability:[20] the most recent release is always available from http://purl.obolibrary.org/obo/duo.owl, while previous versions can be accessed through their date-based PURL, providing choice for users who prefer to use a specific historical view of the ontology[21,22] for stability while transitioning to the latest version.

4. Pilot adoption: once the OWL version of DUO was available, the use of the ontology in live software systems was piloted. This included a pilot by the EGA and Sanger as well as a pilot by the Broad Institutes DUOS data repository[7] were working software systems in both data repositories were referencing the DUO OWL libraries to tag datasets in their system and underlie their search features. DUOS is used in the All-of-Us and ANVIL GA4GH driver projects.

5. GA4GH product approval: Once the use of DUO was demonstrated via GA4GH driver projects pilots, DUO was unanimously approved as a GA4GH standard, following the GA4GH official product review and approval process, by the GA4GH steering committee in Jan 2019.[1]

### Evolution of DUO

Contributions to DUO are public and created by raising GitHub issues,[12] anyone may submit a request to add a new term, or comment on an existing request. Requests are discussed by the DUO work stream and driver project implementers on the tracker, DUO mailing-list and during periodic teleconferences. Once approved, changes are open to the public for further discussion throughout a comment period of two weeks, as per the DUO governance policy.[13] External efforts such as the Informed Consent Ontology (ICO)[11] were additionally reviewed for interoperability and synergistic evolution; DUO has been directly imported in ICO to describe data use conditions instead of duplicating its content.

