## [Document S2. Transparent peer review records for Lawson et al · Cell Genomics]

The Data Use Ontology to streamline responsible access to human biomedical datasets

Jonathan Lawson^{†,1}, Moran N Cabili^{†,1}, Giselle Kerry², Tiffany Boughtwood³, Adrian Thorogood^{4,5}, Pinar Alper⁵, Sarion R. Bowers⁶, Rebecca R. Boyles⁷, Anthony J. Brookes⁸, Matthew Brush⁹, Tony Burdett², Hayley Clissold⁶, Stacey Donnelly¹, Stephanie O.M. Dyke¹⁰, Mallory A. Freeberg², Melissa A Haendel⁹, Chihiro Hata¹¹, Petr Holub¹², Francis Jeanson¹³, Aina Jene¹⁴, Minae Kawashima¹⁵, Shuichi Kawashima¹⁶, Melissa Konopko¹⁷, Irene Kyomugisha¹⁸, Haoyuan Li¹⁹, Mikael Linden²⁰, Laura Lyman Rodriguez²¹, Mizuki Morita²², Nicola Mulder²³, Jean Muller^{24,25}, Satoshi Nagaie²⁶, Jamal Nasir²⁷, Soichi Ogishima²⁶, Vivian Ota Wang²⁸, Laura D. Paglione²⁹, Ravi N. Pandya³⁰, Helen Parkinson², Anthony A. Philipakis¹, Fabian Prasser³¹, Jordi Rambla¹⁴, Kathy Reinold¹, Gregory A. Rushton¹, Andrea Saltzman¹, Gary Saunders¹⁷, Heidi J. Sofia³², John D. Spalding², Morris A. Swertz³³, Ilia Tulchinsky³⁴, Esther J. van Enckevort³³, Susheel Varma³⁵, Craig Voisin³⁴, Natsuko Yamamoto³⁶, Chisato Yamasaki³⁶, Lyndon Zass²³, Jaime M. Guidry Auvil²⁸, Tommi H. Nyrönen²⁰, Mélanie Courtot^{*,2,37}

Summary

Scientific Editor:	Orli Bahcall
Initial submission:	2/28/2021
Revision received:	7/02/2021
Accepted:	8/09/2021
Rounds of review:	2
Number of reviewers:	3

Referee reports, first round of review

Reviewer #1: The paper describes a novel ontology for standardisation of data use agreements for the sharing of global data about human health. This ontology addresses a significant need and has already begun to show its great potential through use in the context of the Global Alliance for Genomics and Health. The article text is brief but well-written and describes the ontology and its application to initial use cases. This ontology together with the accompanying infrastructure supporting more automation in the data access management pipeline will be an invaluable resources for researchers going forward.

The only concern I have with the manuscript in its present form is that I think the presentation of the ontology content could be clarified somewhat. In particular:

1. The Results section seems to miss a comprehensive description of the structure of the ontology and its main design decisions. How many classes does it have? What are the primary types of entity in the ontology and how are they defined? How are they classified within BFO? Which properties are used, how are they defined, how are they used? etc.
2. The first two paragraphs of the Methods do seem to describe the structure of the ontology, albeit very briefly. This seems to fit better within Results.
3. The remainder of the Methods do describe how the ontology is developed, but key details seem to be missing. For example, the GitHub project repository mentions that the ontology development process is based on the "ontology starter kit" but this is not mentioned in the paper. The Japanese language module is mentioned, but what are the other modules and why have those modules been selected?
4. How can people contribute to the ontology?

Reviewer #2:

General impression.

The paper presents a solution to the vexing problem of access difficulty to repository data for secondary analysis caused by a bottleneck at the approval step. Too much data, too many requests and a manual approval creates the bottleneck. The authors make a good case that the automated and ontology-driven solution presented here will be needed to properly scale as the numbers grow.

Several advantages are offered that could immediately reduce the workload (and wait time) for data approval: Once data are annotated (a vary large task, however, not well resolved in this paper), then potential data requestors can learn if they would meet the criteria before submitting a DAR. This will reduce strain on the evaluation system immediately and save the researchers' time spent on fruitless requests.

The authors state: "Data Access Committees (DACs) must manually review data use terms but can be delayed by the need to interpret data use terms described in inconsistent and ambiguous language." DACs will still have that problem for all the data that has not been coded into DUO. No really convincing proposal to resolve that is offered.

It is not clear from the paper how big a job it will be to annotate datasets. There is a large number quoted (>200,000) in the text, but Table 1 shows that they are almost entirely from one location (Japan). Clearly dbGaP and EGA have many that are not encoded with DUO terminology? Who will fund the effort to encode them all and will the effort be burning time and money on datasets that may never be requested? Will it be useful only going forward and then only if the repositories and journals require encoding upon submission?

Specific issues.

author 13: National Institute of Genetics (what country?)
are author 13 and 16 the same place? ELIXIR

.pdf p.7

grant researchers access to multiple datasets based
should be
grant researchers' access to multiple datasets based

p 9.

see accompanying paper by phillipakis, et al. what paper is that? Cabili ... Phillipakis?

p 10.

"A recent standard, Health Level Seven International (HL7) Fast Healthcare Interoperability Resources (FHIR) (30) HL7's Consent resource (31) as well as other tools or standards..."

Is there a comma or something else missing? It is not clear what this sentence means. What exactly is the "recent standard"? One or more the things enumerated thereafter?

The paper touts the ease of interpretation of "DUO adopts the term "not for profit use only"" but it is not really all that clear. Is it missing hyphens in the "not-for-profit" term?

"future work will investigate inclusion of DUO terms in Passport visas, which would allow authenticated researchers to automatically access DAC-approved datasets after sign-in." This would be a major advance, as long as datasets are being properly annotated.

Reviewer #3:

Thank you for the opportunity to review this paper. It presents the DUO from GA4GH, an important component of building datasets that maximise opportunities for genomics research. The standard has already undergone extensive review, and the paper is clearly written. In particular, I found the figures very helpful in driving a clear message of how the standard fits in with data access processes.

My only comment is a fairly broad one: the paper emphasises the application of this standard to research genomics datasets, and it is great to see it has already been implemented by several large research genomics initiatives. In our country, as in many others, the genomics data generated in clinical care now outstrips the data generated by research projects. The biggest challenge going forwards will be to ensure that GA4GH standards are equally applied to datasets generated by diagnostic providers or else we risk vast amounts of data remaining siloed and inaccessible, despite many clinical consent forms including clauses enabling research use. I wonder if the authors would consider re-wording in places to emphasise the importance of implementing these standards not just for the research but also for the clinical and diagnostic communities.

Author response to the first round of review

Editor's comments:

We have now received the final reviewer report on your paper, and a copy of all 3 reports are attached below. We invite you to revise your manuscript in response to these referee comments and the editorial requests below. I will be glad to discuss how to best focus and present this work, in context of this manuscript, and as a part of the special issue.

Author response:

We have reformatted the manuscript and addressed suggestions as described in detail below. Our paper has consequently been substantially improved and we would like to thank the reviewers and editor for their helpful comments.

1) Please revise to present the manuscript in our Technology article format.

Manuscript change:

We have changed our manuscript to fit the "Technology" article format. In particular, we have

- 1) added a "Design" section that outlines the technical development details of DUO,
- 2) added a "Limitations" section to the Discussion section.

2) Please include a STAR Methods section (see details below).

Manuscript change:

We retitled the current Methods section to "STAR Methods" and added "Lead Contact," "Materials Availability," "Data and Code Availability," and "Methods Details" subheadings with appropriate content.

3) You may include additional supporting information in Supplementary Information (see details in email).

Author Response:

We have not included supplementary information as details are provided in the manuscript or STAR Methods section.

Response to Reviewers

Reviewer #1

The paper describes a novel ontology for standardisation of data use agreements for the sharing of global data about human health. This ontology addresses a significant need and has already begun to show its great potential through use in the context of the Global Alliance for Genomics and Health. The article text is brief but well-written and describes the ontology and its application to initial use cases. This ontology together with the accompanying infrastructure supporting more automation in the data access management pipeline will be an invaluable resources for researchers going forward.

Author Response:

We thank the reviewer for their positive comments.

The only concern I have with the manuscript in its present form is that I think the presentation of the ontology content could be clarified somewhat. In particular:

1. The Results section seems to miss a comprehensive description of the structure of the ontology and its main design decisions. How many classes does it have? What are the primary types of entity in the ontology and how are they defined? How are they classified within BFO? Which properties are used, how are they defined, how are they used? Etc.

Manuscript change:

Thank you for the comment. As we wanted to focus on the impact of using DUO we had initially kept the technical description purposely brief. We have now added more details in the Design section on the ontology development itself, in particular regarding the structure of the ontology, including number of classes and organisation under BFO. The primary types of entity in the ontology are described in Figure 1.

2. The first two paragraphs of the Methods do seem to describe the structure of the ontology, albeit very briefly. This seems to fit better within Results.

Manuscript change:

We chose to focus results described in this manuscript around impact of using DUO, and consider DUO and its development as a tool enabling interesting results downstream, which is why its development is described in the Design section. As mentioned above we had therefore purposely kept the technical discussions on DUO to a minimum, but have now added more information about its development in the Design section as well as the STAR methods.

3. The remainder of the Methods do describe how the ontology is developed, but key details seem to be missing. For example, the GitHub project repository mentions that the ontology development process is based on the "ontology starter kit" but this is not

2

mentioned in the paper. The Japanese language module is mentioned, but what are the other modules and why have those modules been selected?

Manuscript change:

Thanks for checking the GitHub repository. We have added more details on the technical development of DUO in design section. We also have added mention of the Ontology Development Kit and citation. The only modules currently considered are language translations from the original english, which are pending validation and conversion into OWL and therefore not described here.

4. How can people contribute to the ontology?

Manuscript change:

Thank you for your interest in contributing to DUO. Paragraph 2 in the Design section includes details for contributing to the ontology as well as criteria for inclusion of new terms. To make this clearer to the reader we have also added a STAR section pointing to the GH repository which is public.

Reviewer #2:

General impression.

The paper presents a solution to the vexing problem of access difficulty to repository data for secondary analysis caused by a bottleneck at the approval step. Too much data, too many requests and a manual approval creates the bottleneck. The authors make a good case that the automated and ontology-driven solution presented here will be needed to properly scale as the numbers grow.

Several advantages are offered that could immediately reduce the workload (and wait time) for data approval: Once data are annotated (a vary large task, however, not well resolved in this paper), then potential data requestors can learn if they would meet the criteria before submitting a DAR. This will reduce strain on the evaluation system immediately and save the researchers' time spent on fruitless requests.

The authors state: "Data Access Committees (DACs) must manually review data use terms but can be delayed by the need to interpret data use terms described in inconsistent and ambiguous language." DACs will still have that problem for all the data that has not been coded into DUO. No really convincing proposal to resolve that is offered.

3

It is not clear from the paper how big a job it will be to annotate datasets. There is a large number quoted (>200,000) in the text, but Table 1 shows that they are almost entirely from one location (Japan). Clearly dbGaP and EGA have many that are not encoded with DUO terminology? Who will fund the effort to encode them all and will the effort be burning time and money on datasets that may never be requested? Will it be useful only going forward and then only if the repositories and journals require encoding upon submission?

Manuscript change:

We thank the reviewer for their comments and have added detail of possible solutions to this problem in the Discussion, limitations section. Some repositories such as EGA are already mandating addition of DUO terms upon submission of new datasets. Indeed, having journal supporting this would help: this information could be added under availability description in manuscripts.

Specific issues.

author 13: National Institute of Genetics (what country?)

are author 13 and 16 the same place? ELIXIR

Manuscript change: We have asked all co-authors to review their affiliations and those have been updated consequently.

.pdf p.7

grant researchers access to multiple datasets based should be

grant researchers' access to multiple datasets based

Manuscript change: We have rephrased as "to help automating access by researchers to

multiple datasets”

4

p 9.

see accompanying paper by phillipakis, et al. what paper is that? Cabili ... Phillipakis?

Manuscript change: Thanks for checking. This is indeed the Cabili et al. manuscript. The different manuscripts will be properly linked and referenced upon publication.

p 10.

"A recent standard, Health Level Seven International (HL7) Fast Healthcare Interoperability

Resources (FHIR) (30) HL7's Consent resource (31) as well as other tools or standards..."

Is there a comma or something else missing? It is not clear what this sentence means.

What exactly is the "recent standard"? One or more the things enumerated thereafter?

Manuscript change: We agree this was hard to parse and have rephrased as “DUO terms can also be used in healthcare settings and alongside complementary standards. Health Level Seven International (HL7) Fast Healthcare Interoperability Resources (FHIR)³⁰ Consent resource³¹ as well as other tools or standards, such as Automatable Discovery and Access Matrix (ADA-M) or OASIS's LegalRuleML³², use logic for expressing more complex data use rules.” Thanks for the helpful comment.

The paper touts the ease of interpretation of "DUO adopts the term "not for profit use only"" but it is not really all that clear. Is it missing hyphens in the "not-for-profit" term?

Manuscript change: Thanks for the comment, we have updated accordingly.

"future work will investigate inclusion of DUO terms in Passport visas, which would allow authenticated researchers to automatically access DAC-approved datasets after sign-in."

This would be a major advance, as long as datasets are being properly annotated.

Reviewer #3:

Thank you for the opportunity to review this paper. It presents the DUO from GA4GH, an important component of building datasets that maximise opportunities for genomics research. The standard has already undergone extensive review, and the paper is clearly written. In particular, I found the figures very helpful in driving a clear message of how the standard fits in with data access processes.

5

My only comment is a fairly broad one: the paper emphasises the application of this standard to research genomics datasets, and it is great to see it has already been implemented by several large research genomics initiatives. In our country, as in many others, the genomics data generated in clinical care now outstrips the data generated by research projects. The biggest challenge going forwards will be to ensure that GA4GH standards are equally applied to datasets generated by diagnostic providers or else we risk vast amounts of data remaining siloed and inaccessible, despite many clinical consent forms including clauses enabling research use. I wonder if the authors would consider re-wording in places to emphasise the importance of implementing these standards not just for the research but also for the clinical and diagnostic communities.

Manuscript change: We thank the reviewer for this excellent point. We had in places mentioned “research and clinical” but this was not consistent. We have updated the manuscript to refer to biomedical data in general, to prevent possible association with only one type of data.

Referee reports, second round of review

Reviewer #1: The authors have addressed all the points I raised before, thus I am pleased to recommend the work for publication.

Reviewer #2: The authors have addressed issues raised by the reviewers.

specific edit comments.

p.10.

"This reflects an inversion of understanding as permitting unique, edge-case like types of research would be permissible via many of the existing DUO terms, particularly those such as General Research Use and Health/Medical/Biomedical Use Ultimately, after engaging with the DUO team, these representatives became strong proponents of the DUO and advocate for its use amongst other rare disease participant groups."

This is difficult to understand. At a minimum, there seems to be a missing period after "Biomedical Use". But the first sentence seems to be two thoughts munged into one. "as permitting" what? and what would "be permissible"? Can't figure it out. Maybe "which" before "would be permissible" would straighten it out?
And shouldn't "edge-case like" be "edge-case-like"?
Also, don't need "st" on "among".

Reviewer #3: Comments enter in this field will be shared with the author; your identity will remain anonymous.

No further comments, the authors have satisfactorily addressed the reviewers' feedback.

Author response to the second round of review

We are pleased to submit a further revised version of our manuscript "The Data Use Ontology to integrate and streamline access to ethically and legally diverse datasets". As a reminder this paper and the papers "Empirical Validation of an Automated Approach to Data Use Oversight" and "GA4GH Passport data access technology standard for distributed genomics & health research" are crossreferenced in the manuscript as paper by "Cabili et al." and "Voisin et al." respectively.

In addition to editorial revisions which we have addressed using track changes, the manuscript has been updated as follow:

- Figure 2 has been updated.
- Descriptive titles and expanded legends have been added to all figures.
- Footnote for table 1 has been removed and included in the legend.
- We have expanded on the type of data access challenges and their impact in the introduction.
- We have significantly expanded on methods for development of the DUO as part of the STAR section.
- We have added a description of the content of DUO as part of the STAR section.
- A description of current implementation in the European Genome-Phenome Archive has been added as part of the STAR section.
- We have clarified how the choice of limiting the scope of DUO is a limitation as it prevents representing all possible combinations of access conditions yet has been successful in representing current use cases.

We confirm that this work is original and has not been published elsewhere, nor is it under consideration for publication elsewhere. Authors have declared conflict of interests in the respective form and manuscript, and a diversity statement is included.

We would like to take this opportunity to thank the reviewers for their helpful contributions in improving our paper, and you and the editorial team as a whole for their help and support in processing the manuscript.

Thank you once again for your consideration of our work.